# The Characteristics and Obstetric Outcomes of Type II Vasa Previa: Systematic Review and Meta-Analysis

**DOI:** 10.3390/biomedicines10123263

**Published:** 2022-12-15

**Authors:** Shinya Matsuzaki, Yutaka Ueda, Satoko Matsuzaki, Mamoru Kakuda, Misooja Lee, Yuki Takemoto, Harue Hayashida, Michihide Maeda, Reisa Kakubari, Tsuyoshi Hisa, Seiji Mabuchi, Shoji Kamiura

**Affiliations:** 1Department of Gynecology, Osaka International Cancer Institute, Osaka 541-8567, Japan; 2Department of Obstetrics and Gynecology, Osaka University Graduate School of Medicine, Osaka 565-0871, Japan; 3Department of Obstetrics and Gynecology, Osaka General Medical Center, Osaka 558-8558, Japan; 4Department of Forensic Medicine, School of Medicine, Kindai University, Osaka 589-8511, Japan

**Keywords:** assisted reproductive technique, vasa previa, bilobed placenta, succenturiate lobes, accessary lobes, type II vasa previa

## Abstract

Vasa previa is a rare fetal life-threatening obstetric disease classified into types I and II. This study aimed to examine the characteristics and obstetric outcomes of type II vasa previa. A systematic review was performed, and 20 studies (1998–2022) were identified. The results from six studies showed that type II vasa previa accounted for 21.3% of vasa previa cases. The characteristics and obstetric outcomes (rate of assisted reproductive technology (ART), antenatal diagnosis, emergent cesarean delivery, maternal transfusion, gestational age at delivery, and neonatal mortality) were compared between type I and II vasa previa, and all outcomes of interest were similar. The association between ART and abnormal placenta (bilobed placenta or succenturiate lobe) was examined in three studies, and the results were as follows: (*i*) increased rate of succenturiate lobes (ART versus non-ART pregnancy; OR (odds ratio) 6.97, 95% confidence interval (CI) 2.45–19.78); (*ii*) similar rate of abnormal placenta (cleavage-stage versus blastocyst embryo transfer); (*iii*) increased rate of abnormal placenta (frozen versus fresh embryo transfer; OR 2.97, 95%CI 1.10–7.96). Although the outcomes of type II vasa previa appear to be similar to those of type I vasa previa, the current evidence is insufficient for a robust conclusion.

## 1. General Overview

The variant of abnormal placenta that is correlated with the most adverse obstetric outcomes is vasa previa [1,2,3,4], which is characterized by unprotected cord vessels running near or across the internal cervical os [1,2,5,6]. The distance between the unprotected cord vessels and internal cervical os used to diagnose vasa previa is controversial, and fetal blood vessels running within 2 cm of the internal cervical os are recognized as vasa previa [7,8,9,10]. The estimated prevalence of vasa previa is approximately 0.05%, and an extremely high fetal mortality rate of 44% has been reported in women with undiagnosed vasa previa [11,12]. In general, vasa previa is classified into two types according to its etiology [13,14,15,16,17,18,19].

The combination of low-lying placenta and velamentous cord insertion has been considered as a high-risk condition of vasa previa [2,20,21], and this condition may be a majority type of vasa previa, classified as type I. In the case of a bilobed placenta or placenta with a succenturiate lobe, unprotected vessels connecting the lobes are often observed [3,14,22,23]. If the unprotected vessels run within 2 cm of the internal cervical os, this condition is classified as type II vasa previa [3,22]. Recent studies have suggested an association between assisted reproductive technology (ART) and type I vasa previa, as women who conceive by ART are more likely to have velamentous cord insertion than those who conceive spontaneously [2,21,24,25]. Type I vasa previa is considered to increase with the increased number of ART pregnancies, whereas the association between ART use and type II vasa previa has not previously been examined.

Unlike type I vasa previa, no previous systematic review has focused on type II vasa previa, and the characteristics and prognosis of this type of vasa previa are understudied due to its rarity. Moreover, no systematic review has compared the characteristics and prognoses of type I and II vasa previa. Therefore, this systematic review aimed to examine the characteristics and prognosis of type II compared to type I vasa previa. This study also aimed to determine the association between ART and abnormal placenta (bilobed placenta or placenta with a succenturiate lobe).

## 2. Systematic Literature Search

### 2.1. Approach for Systematic Review

A systematic literature search was performed to review previous studies on type II vasa previa. This study aimed to determine the following outcomes of primary interest: (*i*) estimated frequency of type II vasa previa, (*ii*) obstetric outcomes of type II vasa previa, and (*iii*) comparison of characteristics between type I and type II vasa previa. The secondary interest of this study was the effect of ART on the prevalence of abnormal placenta (bilobed placenta or placenta with a succenturiate lobe). The influence of ART on the incidence of abnormal placenta was examined according to the type of ART used in the sensitivity analysis. In this study, bilobed placenta and succenturiate lobe caused type II vasa previa; thus, abnormal placenta was defined as bilobed placenta and placenta with succenturiate lobe.

Following the guidelines of the Preferred Reporting Items for Systematic Reviews and Meta-Analyses statement published in 2020 [26], a systematic review was performed using three electronic search engines (PubMed, the Cochrane Central Register of Controlled Trials (CENTRAL), and Scopus) as previously described [27]. Previous studies published before 31 August 2022 were searched and screened using words related to vasa previa and abnormal placenta (Appendix A). In PubMed and Cochrane searches, Medical Subject Headings (MeSH terms) were also used.

### 2.2. Article Retrieval and Search Strategy

Studies were screened by inspecting the titles and abstracts of applicable studies, as previously reported [28]. Titles and abstracts were screened by Shinya Matsuzaki and Misooja Lee. to identify studies that examined the outcomes of interest. The type of study was defined according to the included cases as follows: (*i*) case report (1 or 2), (*ii*) case series (3–10), and (*iii*) original articles (11≥).

### 2.3. Study Selection

To examine the primary and secondary outcomes of interest, the inclusion criteria of studies were as follows: (1) comparative studies comparing an experimental group (type II vasa previa and ART pregnancy) and a control group (type I vasa previa and non-ART pregnancy), (2) the incidence of type II vasa previa was clarified, (3) type II vasa previa was reported in a case report or case series, and (4) the number of women with abnormal placenta and ART was clearly identified. Among the eligible studies, comparative studies that examined the outcomes of interest (sensitivity analysis) were further examined.

The exclusion criteria were as follows: (1) insufficient information to clearly identify the number of patients with type II vasa previa, (2) no abstract or abstract was unavailable, (3) articles not written in English, and (4) conference abstracts, editorials, letters, reviews, systematic reviews, and meta-analyses.

### 2.4. Data Extraction

All the information was extracted by the first author (Shinya Matsuzaki). The first author’s name, year of publication, study location, number of included cases, number of experimental and control groups, and outcomes of interest were recorded. Data included in the analysis were verified by a review author (Misooja Lee).

### 2.5. Analysis of Outcome Measures and Assessment of Bias Risk

The primary aim of the present study was to determine the frequency and obstetric outcomes of type II vasa previa. Since we considered that knowing the differences in type I and II vasa previa may be useful, the characteristics and obstetric outcomes comparing women with type I vasa previa and type II vasa previa were investigated using the included studies as co-primary outcomes. The secondary outcome of the present study was to examine the effect of ART on the prevalence of abnormal placenta and to review the rate of vasa previa in women with abnormal placenta. The influence of ART types on the prevalence of abnormal placenta was determined using sensitivity analysis.

The risk of bias in non-randomized studies of interventions tool (ROBINS-I) was used to assess the risk of bias in included comparator studies, as previously reported [29,30,31,32].

### 2.6. Meta-Analysis Plan

Information was accumulated from the included studies, hazards for the outcomes of interest were calculated using odds ratios (ORs), and 95% confidence intervals (95%CIs) for the outcomes were considered.

The heterogeneity across the studies to evaluate the outcome of interest was examined using the *I*^2^ statistic, which weighs the proportion of the total variation. Fixed or random-effects analysis was performed according to the heterogeneity among the studies, as shown in Appendix A [33]. A meta-analysis was conducted, and images and figures were assembled using RevMan ver. 5.4.1 software (Cochrane Collaboration, Copenhagen, Denmark).

### 2.7. Statistical Analysis

Differences in baseline demographics between the two groups were examined using the chi-squared or Fisher’s exact tests, as appropriate. All statistical analyses were based on two-sided hypotheses, and a *p* value < 0.05 was considered statistically significant. The Statistical Package for Social Sciences (SPSS, version 28.0, IBM, Armonk, NY, USA) or RevMan ver. 5.4.1 software (Cochrane Collaboration, Copenhagen, Denmark) were used for analyses.

## 3. Results

### 3.1. Study Selection

Figure 1 shows the selection scheme used in this study. First, 1364 studies were identified, and 20 studies met the study criteria for descriptive analysis [14,34,35,36,37,38,39,40,41,42,43,44,45,46,47,48,49,50,51,52]. Two studies reported by Suzuki et al. [36,37] used overlapped data (due to overlapping study periods); however, the outcome of interest was different in these studies. The analysis of the prevalence of vasa previa among women with succenturiate lobes was performed in a study published in 2010 [36], and the effect of ART on the prevalence of succenturiate lobe was examined using a study published in 2008 [37].

### 3.2. Study Characteristics

The summary of 20 eligible studies is presented in Table 1. Of these (*n =* 20), 16 studies investigated the vasa previa, and 4 studies concerned abnormal placenta. Regarding the study type (*n =* 20), all studies were retrospective in nature, 14 were original articles, 4 were case series, and 2 were case reports. Of these (*n =* 20), the published years were between 1998 and 2022, and all studies were retrospective [14,34,35,36,37,38,39,40,41,42,43,44,45,46,47,48,49,50,51,52]. No prospective studies or randomized controlled trials have been reported.

Among the eligible studies (*n =* 20), approximately one-third were reported from Europe (*n* = 6, 30.0%) [38,41,44,45,46,51] followed by Japan (*n* = 5, 25.0%) [36,37,39,47,48], the United States (*n* = 5, 25.0%) [14,35,43,49,52], and others (*n* = 4, 20.0%) [34,40,42,50].

### 3.3. Risk of Bias of Included Studies

The risk of bias assessment for the included comparator studies is presented in Appendix A. Among the studies (*n* = 6), a possible moderate publication bias in three studies [40,42,43] and severe publication bias in the other three studies [39,41,44] were observed.

### 3.4. Number of Studies: Primary Outcome

A systematic review was performed to identify previous studies that included women with type II vasa previa, and 16 studies met the criteria (Table 1) [14,38,39,40,41,42,43,44,45,46,47,48,49,50,51,52]. Of these (*n* = 16), seven studies were original articles, six were case series, and the remaining three studies were case reports. Among the case series and reports (*n* = 9), the studies included women with vasa previa, and the outcomes were reported; however, no specific characteristics or outcomes of type II vasa previa have been reported [14,38,45,46,47,48,49,50,51]. Moreover, no studies have compared the characteristics and outcomes of type I and type II vasa previa.

#### 3.4.1. Primary Outcome: The Incidence of Type II Vasa Previa

Seven studies were identified as original articles that examined the characteristics and outcomes of type II vasa previa [39,40,41,42,43,44,52]. Among the original articles (*n* = 7), the rate of type II vasa previa ranged from 7.6% to 37.5% in women with vasa previa. The cumulative rate of type II vasa previa was 21.3% (108/507 cases) among women with vasa previa. Furthermore, the number of women without vasa previa was mentioned in three studies [40,41,42]. In these studies, the prevalence of type II vasa previa in pregnant women was estimated, and the rate ranged from 0.01% to 0.05%.

#### 3.4.2. Co-Primary Outcome: Number of Studies Comparing Type I and II Vasa Previa

Six studies examined the obstetric outcomes of vasa previa, dividing them into type I and type II vasa previa (Table 1 and Table 2). All studies compared some characteristics between type I and type II vasa previa [39,40,41,42,43,44]. 

#### 3.4.3. Co-Primary Outcome: Characteristics and Obstetric Outcomes Comparing Type I and II Vasa Previa

Among the studies that compared the characteristics and obstetric outcomes between type I and II vasa previa (*n* = 6), three studies examined the rate of ART [39,42,44], three determined the rate of antenatal diagnosis [41,42,44], four investigated the rate of emergent cesarean delivery [39,42,43,44], one had information regarding maternal transfusion at delivery [39], and three examined the rate of neonatal death, comparing women with type I and type II vasa previa [39,40,42].

#### 3.4.4. Co-Primary Outcome: The Rate of ART

A meta-analysis was conducted to determine the difference in the rate of ART pregnancy between type I and type II vasa previa (Table 2 and Figure 2). A random-effects analysis was conducted due to the considerable heterogeneity. In the unadjusted pooled analysis (*n* = 3) [39,42,44], women with type I and type II vasa previa had a similar rate of ART pregnancy (Figure 2) (odds ratio (OR) 0.47, 95% confidence interval (95%CI), 0.09–2.45; heterogeneity: *p* = 0.06, *I*^2^ = 64%).

#### 3.4.5. Co-Primary Outcome: The Rate of Antenatal Diagnosis

Since type I and type II vasa previa have different etiologies, we hypothesized that the antenatal diagnosis of type I and type II vasa previa may be different. Specifically, we hypothesized that as type I vasa previa often has a low-lying placenta, the antenatal diagnosis of this type may be higher than that of type II vasa previa. To compare type I and type II vasa previa in the examination of antenatal diagnosis, three studies were included in the meta-analysis [41,42,44]. In this analysis, a fixed-effects analysis was performed because there was no heterogeneity among the studies. In the pooled unadjusted analysis, the rate of antenatal diagnosis was similar between the two groups (Figure 2) (OR 1.32, 95%CI 0.41–4.31; heterogeneity: *p* = 0.51, *I*^2^ = 0%).

#### 3.4.6. Co-Primary Outcome: The Rate of Emergent Cesarean Delivery

To assess the difference in the rate of emergent cesarean delivery, a meta-analysis was conducted, including four eligible studies [39,42,43,44]. Due to low heterogeneity among the studies, a fixed-effects analysis was used. In the unadjusted pooled analysis, the emergent cesarean delivery rate was similar between women with type I and type II vasa previa (Figure 2) (OR 1.00, 95%CI 0.58–1.74; heterogeneity: *p* = 0.27, *I*^2^ = 23%).

#### 3.4.7. Co-Primary Outcome: Gestational Age at Delivery

The gestational age (GA) at delivery was also compared in the present meta-analysis. In the pooled analysis examined by a random-effect analysis due to the considerable heterogeneity among the studies, no significant difference in gestational week at delivery was observed between women with type I and type II vasa previa (Figure 2) (mean difference, −0.05; 95%CI –1.11–1.00; heterogeneity: *p* = 0.04, *I*^2^ = 70%).

#### 3.4.8. Co-Primary Outcome: Maternal and Neonatal Outcomes

Only a limited examination was performed to assess maternal outcomes comparing type I and type II vasa previa. Maternal transfusion rate was examined in one study, and the rate of transfusion at delivery was similar between type I and type II vasa previa (OR 2.18, 95%CI 0.70–6.76).

No previous studies have compared the neonatal transfusion rate between type I and type II vasa previa. Three studies compared the rate of neonatal death between type I and type II vasa previa; however, two studies showed no neonatal death [39,40]. Therefore, a meta-analysis of neonatal death could not be performed. Among the three studies (*n* = 3), two studies showed no neonatal death [39,40] whereas type II vasa previa was associated with an increased rate of neonatal death compared to type I vasa previa (OR 9.20, 95%CI 1.37–61.56) in another study [42].

### 3.5. Number of Studies: Secondary Outcome

Five studies were identified to examine secondary outcomes. Of these (*n* = 5), two studies examined the rate of type II vasa previa in women with abnormal placenta [36,40]. However, no studies have examined the association between ART and the prevalence of type II vasa previa, while one study examined the relationship between ART and abnormal placenta [37].

#### 3.5.1. Secondary Outcome: The Rate of Vasa Previa in Abnormal Placenta

Two studies provided information to estimate the rate of vasa previa in women with abnormal placenta (Table 3) [36,40]. In a Japanese single-institutional retrospective study [36], type II vasa previa was observed in 1/83 (1.2%) women with succenturiate lobes of the placenta (OR 68.5, 95%CI 6.15–762.3; women without succenturiate lobes of the placenta (0.09%)). Moreover, in an Australian single-institutional retrospective study [40], three women (4.4%) with type II vasa previa were found among women with 68 succenturiate or bilobed placentas (OR 15.6, 95%CI 4.48–54.19; women without succenturiate or bilobed placentas (0.3%)).

#### 3.5.2. Secondary Outcome: The Association between ART and Abnormal Placenta

One Japanese single-institutional retrospective study examined the effect of ART on the prevalence of succenturiate lobes of the placenta (Table 4) [37]. In this study (*n* = 7713), ART pregnancy (4/105, 3.8%) was associated with an increased rate of succenturiate lobes of the placenta (OR 6.97, 95%CI 2.45–19.78) compared to those in non-ART pregnancies (43/7608, 0.6%). However, no study has examined the effect of ART on the incidence of type II vasa previa.

### 3.6. Number of Studies: Sensitivity Analysis

Although only a limited number of studies were available to determine the effect of ART on the prevalence of abnormal placenta, ART pregnancies have the potential to increase the rate of abnormal placenta when compared to non-ART pregnancies. A sensitivity analysis was conducted to examine the incidence of abnormal placenta according to ART type. In the sensitivity analysis, two studies were available to examine the outcomes of interest [34,35].

#### The Rate of Abnormal Placenta According to the Type of ART

In a Canadian retrospective cohort study, the effect of embryo stage at transfer on placental histopathology features was examined by comparing cleavage-stage (*n* = 252) and blastocyst (*n* = 425) embryo transfer [34]. In this study, all fresh embryo transfers were performed, and the two different stages at transfer had similar rates of bilobed placenta and succenturiate lobes (OR 1.44, 95%CI 0.50–4.12).

In a US retrospective study, placental pathology was examined by comparing ART pregnancies with fresh or frozen embryo transfers [35]. In this study, frozen embryo transfer was associated with an increased rate of succenturiate lobes of the placenta, both in univariate analysis (OR 2.50, 95%CI 1.08–5.76) and multivariate analysis (OR 2.97, 95%CI 1.10–7.96). Frozen embryo transfer was also associated with a higher incidence of succenturiate lobes of the placenta both in univariate analysis (OR 6.72, 95%CI 1.62–27.77) and multivariate analysis (OR 5.32, 95%CI 1.54–18.38), even when the cases were restricted to day 5 transfers.

## 4. Discussion

### 4.1. Principal Findings

The principal findings of this study are as follows: (*i*) type II vasa previa accounts for approximately 20% of vasa previa, and approximately 4% of abnormal placenta may develop to vasa previa, (*ii*) the characteristics and maternal and neonatal outcomes may be similar between type I and type II vasa previa, (*iii*) ART may be associated with an increased rate of abnormal placenta, and (*iv*) the incidence of abnormal placenta may be higher in frozen embryo transfer than in fresh embryo transfer. This association between ART and a higher incidence of abnormal placenta is interesting. Nevertheless, since the mechanism by which ART causes the development of abnormal placenta has not been determined in a basic study, the mechanisms of the higher incidence of abnormal placenta induced by ART are unresolved.

### 4.2. Strengths and Limitations

The strength of the present study is that it is likely to be the first systematic review that focuses on the characteristics and outcomes of type II vasa previa. Since neonatal morbidity and mortality are high in women with vasa previa, we believe that knowing the characteristics and outcomes of type II vasa previa is useful for clinicians to improve its neonatal outcomes. We also found that ART may be associated with an increased rate of placental abnormalities. While no study has determined the effect of ART on the frequency of type II vasa previa, we consider that ART has the potential to be correlated with a higher occurrence of vasa previa compared to non-ART pregnancies.

Nevertheless, several notable limitations of this study must be recognized. First, all eligible studies were retrospective in nature and included a limited number of cases; therefore, unmeasured bias such as information bias and selection bias may exist. Second, the sample size of type II vasa previa was low in all studies and the presence of a type II error needs to be recognized. Most studies showed only a univariate analysis; therefore, confounding factors could not be excluded. We need to recognize that this study is underpowered to draw robust results regarding the characteristics and outcomes of type II vasa previa. This weakness also needs to be recognized when interpreting the results regarding the association between ART and an abnormal placenta. No studies used multivariate analysis to examine the effect of ART on the incidence of abnormal placenta; therefore, the present study cannot characterize ART as a risk factor for abnormal placenta.

Third, the diagnosis of abnormal placenta may be difficult, and the definition of abnormal placenta was not described in the included studies. Due to the difficulty of diagnosis and the fact that the definition of abnormal placenta may be different among the studies, these points need to be recognized as a strong limitation of this study. Fourth, this systematic review was not pre-registered, as we considered that eligible studies may be limited; therefore, we conducted this study after a preliminary literature search. This point leads to bias in the systematic review and needs to be recognized as a limitation of the present study.

Fifth, the present study may have publication bias and requires careful interpretation. For instance, women with a poor prognosis of type II vasa previa and undiagnosed cases may not be reported. Sixth, recent studies have suggested that fetal blood vessels running within 2 cm of the internal cervical os are considered vasa previa; however, the ideal distance between fetal vessels and internal os is still under debate. The definition of vasa previa may be different among the studies, and this point is a bias of this study when interpreting the results of incidence and outcomes of type II vasa previa.

Finally, the association between the succenturiate lobe of the placenta and ART was examined in a previous study; however, no study has examined the association between bilobed placenta and ART. The effect of ART on the prevalence of type II vasa previa remains unknown.

### 4.3. Comparison with Existing Literature

#### 4.3.1. The Incidence of Type II Vasa Previa

The results of the present systematic review showed that type II vasa previa accounts for approximately 20% of all vasa previa cases. A recent study reported that the incidence of vasa previa may be increasing with the increasing number of ART pregnancies [54]; however, the trends of type II vasa previa are still unknown. If ART pregnancy is associated with an increased prevalence of abnormal placenta; the incidence of type II vasa previa is expected to increase. Further studies are warranted to examine the trends of type II vasa previa.

#### 4.3.2. The Different Characteristics between Type I and Type II Vasa Previa

The present systematic review showed that a limited number of studies comparing type I and type II vasa previa are available. From the available data, the rate of ART, antenatal diagnosis, GA at delivery, emergent cesarean delivery, and maternal transfusion appeared to be similar. With regard to neonatal mortality, two of three studies [39,40] had no neonatal death in either type I or type II vasa previa, whereas one study [42] showed significantly higher neonatal mortality in women with type II vasa previa (OR 9.20, 95%CI 1.37–61.56) than in those with type I vasa previa. This point was not discussed in a previous study [42], and further studies are required to determine the neonatal mortality in women with type II vasa previa.

#### 4.3.3. The Association between ART and Abnormal Placenta

With regard to vasa previa, previous studies have reported an association between ART and an increased rate of vasa previa [2,21,24,25,55,56,57,58]. However, these were not classified as type I or type II vasa previa. The association between ART pregnancy and an increased rate of velamentous cord insertion has been reported; therefore, type I vasa previa was considered to have increased in the past decade [24,59,60,61,62]. Unlike type I vasa previa, the association between ART and type II vasa previa has not been focused on in previous studies. To address this problem, the present study examined the association between ART and abnormal placenta and revealed that this association remains understudied.

A study reported by Jauniaux E et al. in 1990 [59] was not included in this study since the study could not be identified by the search terms of the present study; however, this study also reported that ART pregnancy was associated with an increased rate of abnormal placenta (11/50 (22%), *p* < 0.05) compared to those in non-ART pregnancies (3/50 (6%)). While the available data are limited, ART has the potential to increase the rate of abnormal placenta, and frozen embryo transfer appears to be associated with an increased rate of abnormal placenta.

#### 4.3.4. The Mechanism of Developing Type II Vasa Previa

The mechanism of developing bilobed placenta and succenturiate lobe is believed to result from localized atrophy due to poor decidualization [22,23,37]. These changes may be more likely to be observed in the lower uterine segment due to the poor blood supply, and lead to type II vasa previa [63,64,65,66]. However, these statements are based on a hypothesis, and the mechanisms of bilobed placenta, succenturiate lobe, and type II vasa previa are still understudied [67]. Moreover, our systematic literature search revealed that no basic research has examined the bilobed placenta and succenturiate lobe, focusing on the mechanism of the development of type II vasa previa.

Unlike bilobed placenta and succenturiate lobe, the development of velamentous cord insertion has been well discussed, and two major hypotheses have been proposed (trophotropism hypothesis and polarity hypothesis) [54,62]. The trophotropism hypothesis was proposed to explain the development of velamentous cord insertion. The placenta in early pregnancy migrates with proceeding gestational weeks for a better blood supply from a more vascularized area, and these migrations may lead to an unprotected cord with velamentous cord insertion [68,69,70]. In our previous meta-analysis [54], ART was associated with an increased rate of velamentous cord insertion. These results appear to support the trophotropism theory, because ART disrupts the precise timeline of biological action required for blastocyst implantation at multiple stages [62,71].

Since the mechanism of abnormal placenta development is understudied, the association between ART and abnormal placenta is difficult to discuss, and further studies are warranted to resolve the mechanism of abnormal placenta development. The risk of obstetrical complications such as placenta accreta spectrum is different among the types of ART [72], and the assessment of the prevalence of abnormal placenta according to the type of ART may help research to identify the mechanism of development of abnormal placenta.

#### 4.3.5. The Association between ART and Vasa Previa

Although ART has been reported as a risk factor for vasa previa [2,21,24,25,58], our previous study found only one comparative study compared the rate of vasa previa between women who conceived by ART and without ART [73], and the study did not consider the differences in the type of vasa previa. The majority of type I vasa previa consists of a combination of a low-lying placenta and velamentous cord insertion [1,2,5]. Since ART pregnancy is associated with an increased rate of both velamentous cord insertion [59,74,75,76,77,78] and abnormal placentation, including low-lying placenta [75,79,80,81,82,83], ART may be associated with a higher incidence of type I vasa previa.

Although the association between ART and abnormal placenta remains understudied, we consider that ART has the potential to be associated with a higher incidence of type II vasa previa for the following reasons: (*i*) a previous study suggested an association between ART use and increased rate of abnormal placentation, (*ii*) frozen embryo transfer is associated with an increased rate of prevalence of abnormal placenta, and (*iii*) the rate of ART pregnancy was similar between types I and II in the present study.

Although specific data regarding ART use and vasa previa are scarce, we hypothesized that ART use is associated with increased rates of both type I and type II vasa previa. Based on the results of our previous systematic review [54] and the present study, a possible mechanism for the increased rate of type I and type II vasa previa in ART pregnancies is presented in Figure 3.

## 5. Conclusions and Implications

### 5.1. Implications for Practice

While the currently available evidence for type II vasa previa is scarce, the characteristics and outcomes appear to be similar with type I vasa previa. Despite this is only our opinion, one study has shown worse neonatal mortality in women with type II vasa previa; therefore, clinicians may consider treating type II vasa previa as a high-risk type of vasa previa. Elucidation of the cause of worse prognosis may contribute to a reduction in neonatal death in type II vasa previa.

The present study has shown that approximately 1–4% of abnormal placentas are complicated with type II vasa previa, and clinicians need to notify the presence of vasa previa in such cases.

### 5.2. Implications for Clinical Research

Type II vasa previa is a rare disease and no systematic review has focused on this subtype of vasa previa. Based on the limited available data, type II vasa previa has similar characteristics and obstetric outcomes as type I vasa previa. Future studies that focus on type II vasa previa are warranted to consider optimal management according to the type of vasa previa.

Current evidence on the association between ART and abnormal placenta remains scarce. However, a prospective study and randomized controlled study may be difficult to conduct, and a large-scale retrospective study may therefore be suitable to examine the relationship between ART and abnormal placentation. To exclude confounding factors, multivariate analysis, propensity score matching, and inverse probability of treatment weighting are warranted.

## Figures and Tables

**Figure 1 biomedicines-10-03263-f001:**
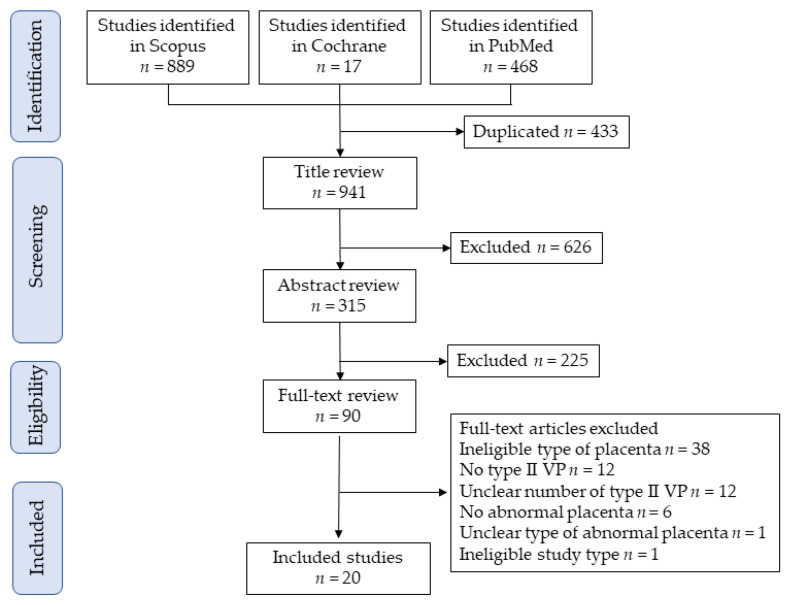
Study selection scheme of the systematic search of previous studies. Abbreviation: VP, vasa previa.

**Figure 2 biomedicines-10-03263-f002:**
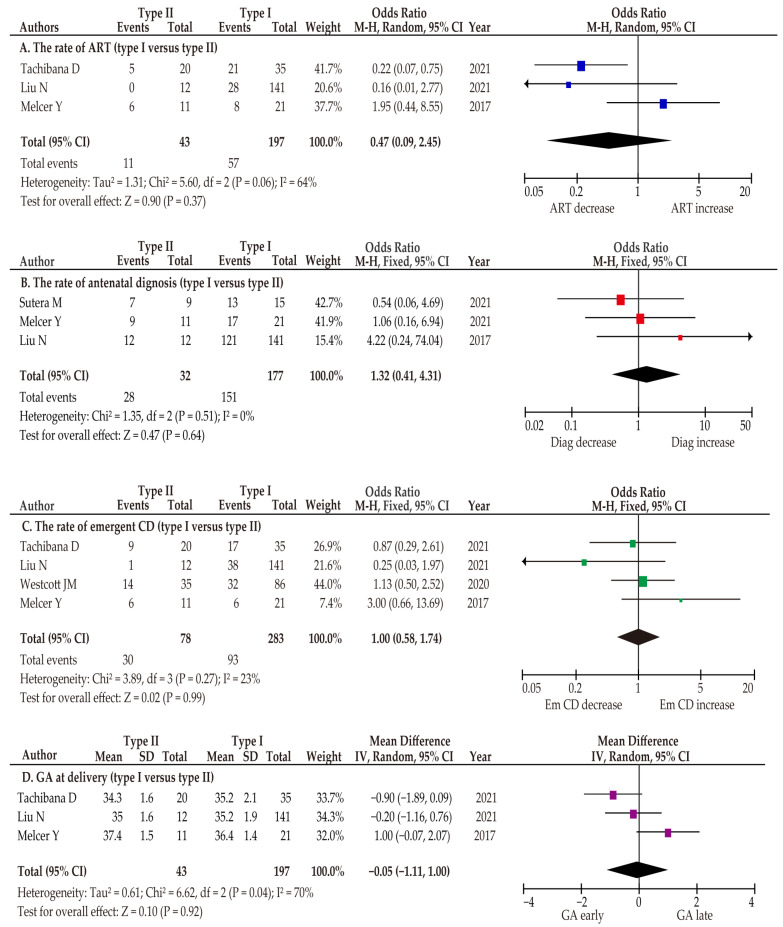
Meta analysis of the characteristics and outcomes of type I and type II vasa previa. The pooled odds ratios for (**A**) the rate of ART (unadjusted), (**B**) the rate of antenatal diagnosis (unadjusted), (**C**) the rate of emergent cesarean delivery (unadjusted), and (**D**) gestational age at delivery between women with type I and type II vasa previa are shown. Forest plots were ordered by the year of publication and relative weight (%) of the study within the strata. The heterogeneity among the studies in each analysis was as follows: substantial heterogeneity (**A**: *I*^2^ = 64%) in the unadjusted analysis, no heterogeneity (**B**: *I*^2^ = 0%) in the unadjusted analysis, low heterogeneity (**C**: *I*^2^ = 23%) in the unadjusted analysis, and substantial heterogeneity (**D**: *I*^2^ = 70%) in the unadjusted analysis. The above results were calculated using RevMan ver. 5.4.1 and may differ slightly from the original values. Abbreviations: CI, confidence interval; ART, assisted reproductive technology; Emergent CD, emergent cesarean delivery; GA, gestational age.

**Figure 3 biomedicines-10-03263-f003:**
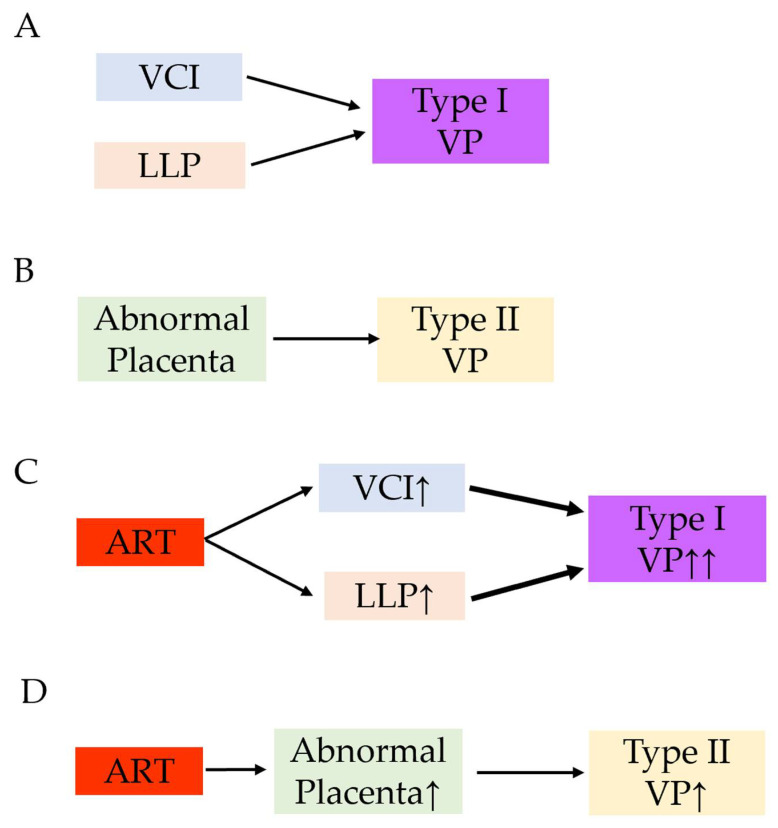
A hypothesis of the increased rate of type I and type II vasa previa in ART pregnancy. Reproduced and updated the data from *Biomedicines*. 2022 Jul 17; 10 (7): 1722. Matsuzaki S et al. [54]. (**A**) The combination of velamentous cord insertion and low-lying placenta is a high-risk condition of vasa previa. (**B**) Abnormal placenta such as bilobed placenta or the placenta with a succenturiate lobe is a risk factor of type II vasa previa. (**C**) Women with ART are more likely to have velamentous cord insertion and low-lying placenta compared to those without ART pregnancy. Increased rate of velamentous cord insertion and low-lying placenta may lead to the increased rate of type I vasa previa. (**D**) ART may be associated with the higher incidence of abnormal placenta. This may lead to the increased rate of type II vasa previa. Abbreviations: VP, vasa previa; ART, assisted reproductive technology; VCI, velamentous cord insertion; LLP, low-lying placenta; ↑, increase; ↑↑, markedly increase.

**Table 1 biomedicines-10-03263-t001:** Previous studies examining women with type II vasa previa.

Author	Year	Location	Type	Total	VP	N	Frequency	ART	Outcome	Type I vs. II
Degirmenci Y [38]	2022	GER	Series	--	5	3	--	--	Yes	--
Tachibana D [39]	2021	JPN	Original	--	55	13	13/55 (23.6%)	Yes	Yes	Yes
Gross A [40]	2021	AUS	Original	5905	21	3	3/21 (14.3%)	--	--	Yes
Sutera M [41]	2021	ITA	Original	89,600	24	9	9/24 (37.5%)	--	--	Yes
Liu N [42]	2021	CHN	Original	79,647	157	12	12/157 (7.6%)	Yes	Yes	Yes
Westcott JM [43]	2020	USA	Original	--	122	35	35/122 (28.7%)	--	--	Yes
Melcer Y [44]	2017	GBR	Original	--	32	11	11/32 (34.4%)	Yes	Yes	Yes
Nohuz E [45]	2017	FRN	Series	--	8	5	--	--	--	--
Catanzarite V [52]	2016	USA	Original	--	96	25	25/96 (26.0%)	--	--	--
Carnide C [46]	2012	PRT	Case	--	1	1	--	--	Yes	--
Kikuchi A [47]	2011	JPN	Case	--	1	1	--	--	Yes	--
Hasegawa J [48]	2010	JPN	Series	--	10	4	--	--	Yes	--
Chmait RH [49]	2010	USA	Case	--	2	2	--	--	Yes	--
Catanzarite V [14]	2001	USA	Series	--	10	8	--	--	Yes	--
Fung TY [50]	1998	HKG	Series	--	3	1	--	--	Yes	--
Baschat AA [51]	1998	GER	Series	--	5	4	--	--	Yes	--

Data are presented as numbers (percentage per column). Abbreviations: GER, Germany; JPN, Japan; AUS, Australia; ITA, Italy; CHN, China; USA, United States of America; GBR, United Kingdom; FRN, France; PRT, Portugal; HKG, Hong Kong; Type, type of study; N, number of included cases; VP, vasa previa; ART, assisted reproductive technology pregnancy; Original, original article; Series, case series; Case, case report; vs., versus; Outcome, obstetric outcomes were clarified; --, not applicable.

**Table 2 biomedicines-10-03263-t002:** The comparison of characteristics and outcomes between type I and type II vasa previa.

Author	Year	Total	N	Type I	Type II	OR (95%CI)
ART						
Tachibana D [39]	2021	55	20 ^#^	21/35 (60.0%)	5/20 (25.0%)	0.22 (0.07–0.75)
Liu N [42]	2021	157	12	28/141 (19.9%)	0/12 (0%)	0.16 (0.01–2.77)
Melcer Y [44]	2017	32	11	8/21 (38.1%)	6/11 (54.5%)	1.95 (0.09–2.45)
Antenatal Diagnosis						
Sutera M [41]	2021	24	9	13/15 (86.7%)	7/9 (77.8%)	0.54 (0.06–4.69)
Liu N [42]	2021	157	12	121/141 (85.8%)	12/12 (100%)	1.06 (0.16–6.94)
Melcer Y [44]	2017	32	11	17/21 (81.0%)	9/11 (81.8%)	4.22 (0.24–74.04)
Emergent cesarean delivery						
Tachibana D [39]	2021	55	20 ^#^	17/35 (48.6%)	9/20 (45.0%)	0.87 (0.29–2.61)
Liu N [42]	2021	157	12	38/141 (27.0%)	1/12 (8.3%)	0.25 (0.03–1.97)
Westcott JM [43]	2020	122	35	32/86 (37.2%)	14/35 (40%)	1.81 (0.84–3.91)
Melcer Y [44]	2017	32	11	6/21 (28.6%)	6/11 (54.5%)	3.00 (0.66–13.69)
GA at delivery						
Tachibana D [39]	2021	55	20 ^#^	35.2 ± 2.1 ^$^	34.3 ± 1.6	0.90 (−0.09–1.89)
Liu N [42]	2021	157	12	35.2 ± 1.9	35.0 ± 1.6	0.20 (−0.76–1.16)
Melcer Y [44]	2017	32	11	36.4 ± 1.4	37.4 ± 1.5	−1.00 (−2.07–0.07)
Maternal Transfusion						
Tachibana D [39]	2021	55	20 ^#^	11/35 (31.4%)	10/20 (50.0%)	2.18 (0.70–6.76)
Neonatal Death						
Tachibana D [39]	2021	55	20 ^#^	0/35 (0%)	0/20 (0%)	--
Liu N [42]	2021	157	12	3/141 (2.1%)	2/12 (16.7%)	9.20 (1.37–61.56)
Gross A [40]	2021	21	3	0/18 (0%)	0/3 (0%)	--

Data are presented as means ± standard deviation or numbers (percentage per column). ^$^ In this study, continuous data are shown as median and range values. The standard deviation was estimated using statistical algorithms reported by Hozo et al. [53]. ^#^ Seven cases were vasa previa with vessels branching out from the placental surface and returning to the placental cotyledons (type III vasa previa). Abbreviations: Total, total number of included cases; N, number of included cases; Type I, type I vasa previa; Type II, type II vasa previa; ART, assisted reproductive technology; OR, odds ratio; CI, confidence interval.

**Table 3 biomedicines-10-03263-t003:** The examination for secondary outcomes and sensitivity analysis.

**Author**	**Year**	**N**	**VP**	**Placenta**
The rate of vasa previa in women with succenturiate lobes				
Gross A [40]	2021	68	3	Succenturiate lobes, bilobed
Suzuki S [36]	2010	83	1	Succenturiate lobes
**Author**	**Year**	**N**	**Ab**	**Type of ART**
ART and abnormal placenta				
Volodarsky A [34]	2021	677	17	Blastocyst versus Cleavage
Sacha CR [35]	2020	1140	70	Frozen versus Fresh
Suzuki S [37]	2008	7713	47	ART versus non-ART

The number of values for each examination is shown. Abbreviations: N, number of included cases; VP, vasa previa; ART, assisted reproductive technology; Placenta, type of abnormal placenta; Ab, number of abnormal placenta; bilobed, bilobed placenta.

**Table 4 biomedicines-10-03263-t004:** The association between ART and abnormal placenta.

Author	Year	Total	N	Placenta	Exp	Cont	Unadjusted OR (95%CI)	Adjusted OR (95%CI)
ART vs. non-ART								
Suzuki S [37]	2008	7713	47	Suc	4/105 (3.8%)	43/7608 (0.6%)	6.97 (2.45–19.78)	--
Blastocyst vs. Cleavage (Fresh)								
Volodarsky A [34]	2021	677	17	Bil, Suc	12/425 (3.3%)	5/252 (2.0%)	1.44 (0.50–4.12)	--
Frozen vs. Fresh								
Sacha CR [35]	2020	1140	70	Suc	22/211 (10.4%)	48/929 (5.2%)	2.50 (1.08–5.76)	2.97 (1.10–7.96)
Day 5 Frozen vs. Fresh								
Sacha CR [35]	2020	630	39	Suc	20/184 (10.9%)	19/466 (4.1%)	6.72 (1.62–27.77)	5.32 (1.54–18.38)

The number of values for each examination is shown. Abbreviations: Total, total number of included cases; N, number of abnormal placenta cases; Frozen, frozen embryo transfer; Fresh, fresh embryo transfer; vs., versus; ART, assisted reproductive technology; Placenta, type of abnormal placenta; Exp, experimental group; Cont, control group; OR, odds ratio; CI, confidence interval; Suc, succenturiate lobes; Bil, bilobed placenta; and --, not applicable.

## Data Availability

All the studies used in this study are published in the literature.

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
