# Peer review of "The Characteristics and Obstetric Outcomes of Type II Vasa Previa: Systematic Review and Meta-Analysis"

_biomedicines, 2022, doi:10.3390/biomedicines10123263_

Round 1

Reviewer 1 Report

The authors of the manuscript „the characteristics and obstretic outcomes of type II vasa previa“ provided a systematic review and metaanalysis on this subject. The statistics are correct and the article concept is sound.

Author Response

Reviewer #1

The authors of the manuscript „the characteristics and obstretic outcomes of type II vasa previa“ provided a systematic review and metaanalysis on this subject. The statistics are correct and the article concept is sound.

Reply:

We sincerely appreciate the reviewer’s positive comments. We trust that the revised manuscript will now be suitable for publication in Biomedicines.

Reviewer 2 Report

Comments about the manuscript:

“The characteristics and obstetric outcomes of type II vasa previa: systematic review and meta-analysis”

Vasa previa is an obstetric disease divided into types I and II involving the vital prognosis of the foetus. No systematic review of the bibliography has yet been established to compare the characteristics of types I and II. The work presented here is a systematic review of various bibliographic data, published in English and from several countries. The purpose of this meta-analysis was to examine the characteristics of type II versus type I and to determine the association between assisted reproductive technology (ART) and the presence of a placenta with a bilobate or succenturiate lobe.

This work seems to me useful and well carried out and seems to me to be suitable for publication. I will just have a few minor remarks regarding the tables.

Page 8, figure 2. Check if the abbreviations of the legend are indeed found in the table. For example, I couldn't find the abbreviations SE and VP in the table, "antenatal diagnosis is spelled out in the table and not abbreviated as "Diag", the table says "emerging CD" and not "Em CD".

Page 10, table 3: same comment. “and --, not applicable.”: I did not find this mention in the table.

Page 10, table 4: same comment. I didn’t find “OR” and “CI” in the table.

The legends of all tables and figures need to be checked carefully.

Check whether the bibliographical references are presented in a homogeneous manner and in accordance with the standards of the journal.

Author Response

Reviewer #2

Comments about the manuscript:

The characteristics and obstetric outcomes of type II vasa previa: systematic review and meta-analysis”

Vasa previa is an obstetric disease divided into types I and II involving the vital prognosis of the foetus. No systematic review of the bibliography has yet been established to compare the characteristics of types I and II. The work presented here is a systematic review of various bibliographic data, published in English and from several countries. The purpose of this meta-analysis was to examine the characteristics of type II versus type I and to determine the association between assisted reproductive technology (ART) and the presence of a placenta with a bilobate or succenturiate lobe.

This work seems to me useful and well carried out and seems to me to be suitable for publication. I will just have a few minor remarks regarding the tables.

Reply:

Thank you for your positive comments. The authors would like to thank the reviewer for his/her constructive critique to improve the manuscript. We have made every effort to address the issues raised and to respond to all comments. Please, find next a detailed, point-by-point response to the reviewer's comments. We hope that our revisions would meet the reviewer’s expectations.

Reviewer #2, comment #1

Page 8, figure 2. Check if the abbreviations of the legend are indeed found in the table. For example, I couldn't find the abbreviations SE and VP in the table, "antenatal diagnosis is spelled out in the table and not abbreviated as "Diag", the table says "emerging CD" and not "Em CD".

Reply: Figure 2

Thank you for your helpful comments. Per the reviewer’s suggestion, we have revised the Figure legend.

Reviewer #2, comment #2

Page 10, table 3: same comment. “and --, not applicable.”: I did not find this mention in the table.

Reply: Table 3

We would like to thank the reviewer for the comments. As the reviewer suggested, we have removed the unnecessary citations.

Reviewer #2, comment #3

Page 10, table 4: same comment. I didn’t find “OR” and “CI” in the table.

The legends of all tables and figures need to be checked carefully.

Reply: Table 4

We respectfully acknowledge the reviewer’s comment and appreciate the reviewer for pointing this out. According to the reviewer’s suggestion, we re-checked the tables carefully. During this process, we have noticed that we missed one eligible study that met the inclusion of this study (Obstet Gynecol. 2016;128:1153-1161). We have added this study and revised the main text accordingly.

Reviewer #2, comment #4

Check whether the bibliographical references are presented in a homogeneous manner and in accordance with the standards of the journal.

Reply: References

We appreciate the reviewer’s comment. Please also refer the reply to the Editor, comment #1. We have modified the references according to the reviewer’s comments.